# The number of neurons in *Drosophila* and mosquito brains

**Joshua I. Raji, Christopher J. Potter** *

The Solomon H. Snyder Department of Neuroscience, The Center for Sensory Biology, Johns Hopkins University School of Medicine, Baltimore, MD, United States of America

* cpotter@jhmi.edu

## Abstract

Various insect species serve as valuable model systems for investigating the cellular and molecular mechanisms by which a brain controls sophisticated behaviors. In particular, the nervous system of *Drosophila melanogaster* has been extensively studied, yet experiments aimed at determining the number of neurons in the *Drosophila* brain are surprisingly lacking. Using isotropic fractionator coupled with immunohistochemistry, we counted the total number of neuronal and non-neuronal cells in the whole brain, central brain, and optic lobe of *Drosophila melanogaster*. For comparison, we also counted neuronal populations in three divergent mosquito species: *Aedes aegypti*, *Anopheles coluzzii* and *Culex quinquefasciatus*. The average number of neurons in a whole adult brain was determined to be 199,380 ±3,400 cells in *D. melanogaster*, 217,910 ±6,180 cells in *Ae. aegypti*, 223,020 ± 4,650 cells in *An. coluzzii* and 225,911±7,220 cells in *C. quinquefasciatus*. The mean neuronal cell count in the central brain vs. optic lobes for *D. melanogaster* (101,140 ±3,650 vs. 107,270 ± 2,720), *Ae. aegypti* (109,140 ± 3,550 vs. 112,000 ± 4,280), *An. coluzzii* (105,130 ± 3,670 vs. 107,140 ± 3,090), and *C. quinquefasciatus* (108,530 ±7,990 vs. 110,670 ± 3,950) was also estimated. Each insect brain was comprised of 89% ± 2% neurons out of its total cell population. Isotropic fractionation analyses did not identify obvious sexual dimorphism in the neuronal and non-neuronal cell population of these insects. Our study provides experimental evidence for the total number of neurons in *Drosophila* and mosquito brains.

## Introduction

Dipteran insects such as the vinegar fly and mosquito can perform startlingly complex tasks that include foraging, courtship, learning, and escape from predators. Understanding how the brains of these insects perform such tasks can highlight common mechanisms of neuronal processing. Towards this goal, quantifying the total number of cells in the brain provides some insight regarding the potential processing brain power of the insect. The cellular composition of the brain can also inform our understanding of evolutionary development, comparative neuroanatomy, and pathophysiological issues in neuroscience [1]. For example, by comparing populations of brain cells, information on aging [2], neurodegenerative diseases [3], evolution [1, 4], sexual dimorphisms [5], and cognitive performance [4, 6] can be obtained in animals.

**Data Availability Statement:** All relevant data are within the manuscript and its Supporting Information files.

**Funding:** This work was supported by grants from the United States Department of Defense to C.J.P. (W81XWH-17-PRMRP), from the National Institute

of Allergy and Infectious Diseases to C.J.P. (NIAID R01AI137078), and a Johns Hopkins Malaria Research Institute Postdoctoral Fellowship to J.I.R.

**Competing interests:** The authors have declared that no competing interests exist.

The complete neural circuitry of the *Drosophila* brain is not yet known, but efforts are underway for its complete reconstruction. Because quantitative mapping of cell type and synaptic density distributions across the insect brain is challenging and difficult to interpret [7], we focus here on estimating the number of neurons in the brain.

The number of neurons in the brain can be estimated using stereological methods such as the optical dissector and fractionator [8, 9]. Stereological counting is applied to heterogeneous brain structures by subdividing the structure into smaller, more uniform components. However, this approach is prone to error because estimates are obtained from cell densities and based on the homogeneity of nuclei in the sample [10–12]. Minimizing this error would require the burden of subdividing the brain into smaller parts with homogenous cell density. Also, given stereological method derives estimates by multiplying cell density and volume, the results are not independent variables and are considered less reliable to use in statistical comparisons against volume [12].

The isotropic fractionator (IF) method involves the combination of direct enumeration and immunohistochemistry, and has the advantage of circumventing the drawbacks in stereological methods [10–12]. The IF method involves the conversion of non-uniform tissue structure into a homogenous single-cell suspension [13] (**Fig 1**). This is achieved by completely dissociating a fixed tissue into a uniform nuclear suspension of known volume. The total cell, neuronal, and glial cell population can be estimated by staining the single-cell suspension and counting cells in defined volumes on a Neubauer chamber. The results obtained from manual counts of nuclei prepared by IF method is reproducible using modern flow cytometry [14, 15].

An estimate of brain cell population using IF has been obtained for several mammalian species including insectivorous mammals [16], rodents [17], cetaceans [18], fish [19], non-human primates [14, 20], and humans [21]. Unlike the complex mammalian brain, the insect brain is primarily composed of neuronal cells [22, 23]. Non-neuronal cells located in the brain are primarily glial, but may also contain other cell types including hemocytes [22]. Although the brain cell population is well studied in mammals, the total number of cells in insect brains remains largely unknown. The closest effort to estimate the number of neurons in the *Drosophila* brain was limited to the EM-reconstructed hemibrain–a part that constitutes approximately a third of the *Drosophila* central brain [7]. The analysis of the brain connectome revealed approximately 25,000 neurons in this small region of the central brain [7]. The total cell population of the *Drosophila* first instar larval brain has been estimated to approximately 9,000 cells by dissociating the brains into single cell suspension and counted on a hemocytometer [24].

The number of neuronal cells currently reported for *Drosophila* and mosquito brains is, to our knowledge, anecdotal. To address this gap, we use the IF method to determine the number of total cells, neuronal, and non-neuronal cells in the whole brain, central brain, and optic lobes of *Drosophila melanogaster*, *Aedes aegypti*, *Anopheles coluzzii*, and *Culex quinquefasciatus*. We also investigated if there is sexual dimorphism in the neuronal and non-neuronal cell population in these insects' brains and dissected brain regions.

## Materials and methods

### Insect brain preparation and dissection

The brains of adult *D. melanogaster*, *Ae. aegypti*, *An. coluzzii*, and *C. quinquefasciatus* (**Fig 1A**), aged 4-6-days post-eclosion were used in the study. Brains were dissected into ice-cold phosphate buffered saline (0.1M PBS, pH 7.2). Followed by fixation for 3 h at 4°C in buffered 4% paraformaldehyde containing 0.1 M sodium phosphate buffer (Millonig's pH 7.4) and

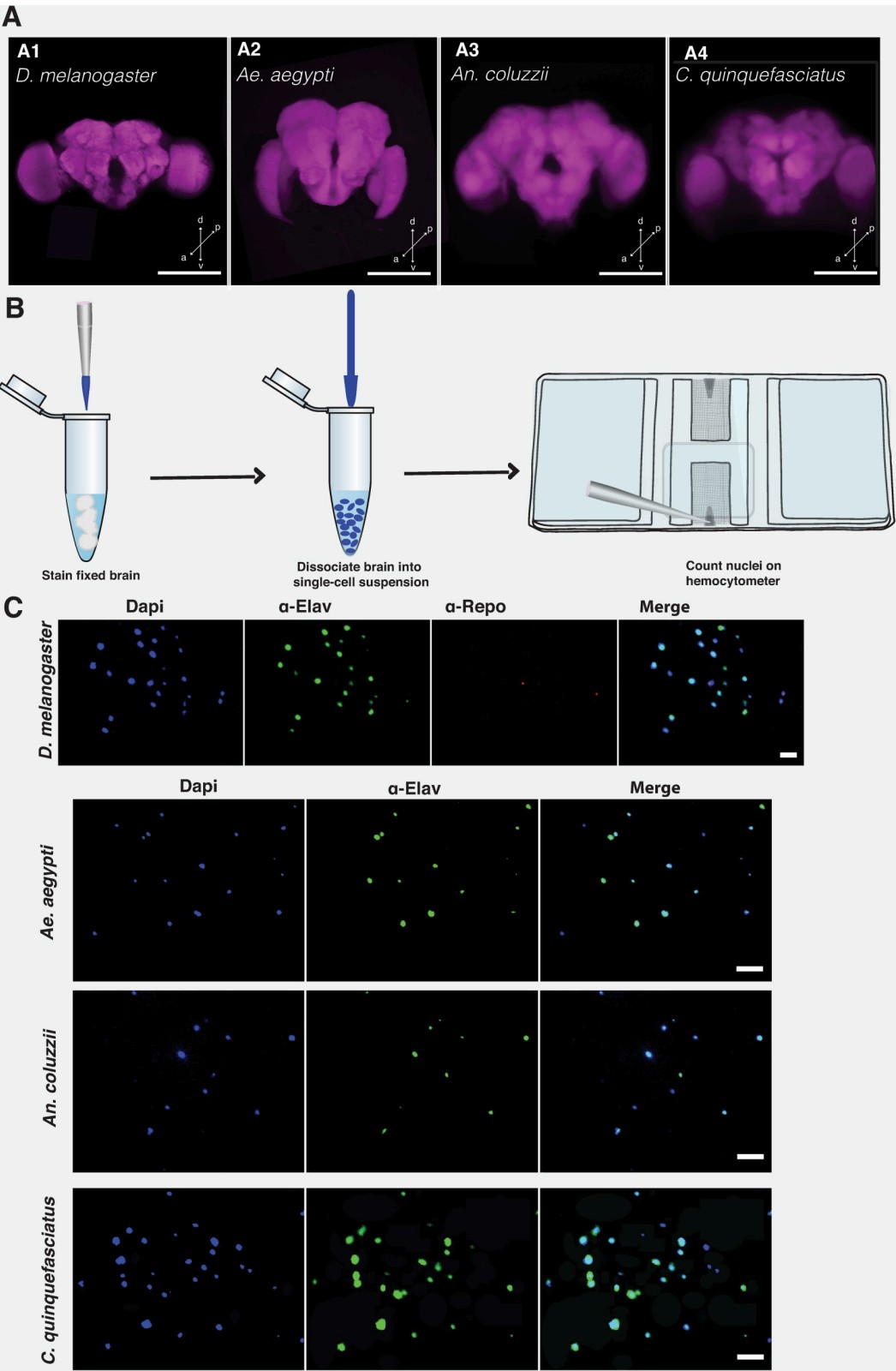

**Fig 1. Estimating brain cell population using isotropic fractionator method.** Confocal Z-stacks of the whole brain of *D. melanogaster* (A1), *Ae. aegypti* (A2), *An. coluzzii* (A3), and *C. quinquefasciatus* (A4). Brain was counterstained with nc82 antibody that labels the pre-synaptic active zone protein, Bruchpilot. Whole brain was imaged on a LSM 700 Zeiss confocal

microscope at 512 × 512 pixel resolution, with 1.04 μm pixel size. Scale bar represents 50 μm. Arrow directions represent the dorsal (d), ventral (v), posterior (p), and anterior (a), respectively. (B). Cartoon showing the steps for performing isotropic fractionator. Brain tissue is fixed and stained with the appropriate dye or antibody. Tissue is dissociated into a single-cell suspension, which is then counted on a Neubauer counting chamber or hemocytometer. (C) Images of nuclei are stained with DAPI (4', 6-diamidino-2-phenylindole dihydrochloride), α-elav (embryonic lethal abnormal vision), and α-repo (Reversed polarity protein) to label nuclei, neuronal and non-neuronal cells, respectively. Staining of glial cells with α-repo was possible only for *Drosophila* brain cells. Scale bar represents 10 μm. For illustration purposes, all images were processed on ImageJ software and Adobe Illustrator.

0.25% Triton X-100. The optic lobes and the central brain were dissected using fine forceps (Dumont #5) under a stereo microscope.

## Immunostaining

For the cell count experiment, quick rinse of the brains was performed 3 times post-fixation in PBT (1X PBS with 0.3% Triton X-100) followed by a prolonged wash for 30 mins on a nutator. Tissue was blocked at room temperature in blocking solution (PBT mix containing 5% Normal Goat Serum) for at least 30 mins. After blocking, tissue samples were stained with primary antibody diluted in blocking solution and incubated at 4˚C for 2 days on a nutator. Washing was performed three times in PBT at RT (at least 20 minutes per wash) after staining with primary antibodies. Thereafter, brains were incubated in secondary antibody diluted in blocking solution at 4˚C for 2 days.

For *Drosophila*, neuronal cells and glial cells were stained with rat anti-elav (DSHB, 1:250) and mouse anti-Repo (DSHB, 1:500) primary antibodies, respectively. These antibodies have been established for staining neurons and glial cells in *Drosophila* [25]. Mosquito brain neurons were also stained with rat anti-elav (DSHB, 1:250). Elav antibody has been used to stain neurons in the central and peripheral nervous system of *Anopheles* mosquito larvae [26]. To estimate total cell population, brain cells for all the four insects were stained in a non-intercalating fluorescent nucleic acid dye solution containing DAPI (4',6-diamidino-2-phenylindole dihydrochloride) suspended in 1X PBS (1:1000) for at least 30 min at room temperature. Secondary antibodies used are Alexa 488 goat anti-rat (Invitrogen 1:250) and Cy3 goat anti-mouse (Jackson ImmunoResearch 1:250). Variable staining was found with anti-Repo on mosquito brain cells. Given this drawback, we estimated non-neuronal cell population in mosquito brains by subtracting elav-positive neuronal cells from DAPI-stained cells.

Whole brain imaging (**Fig 1A**) was performed as described previously (Riabinina et al., 2016). Briefly, adult insect brains were dissected in PBT and washed at room temperature for 1 h. Brains were permeabilized at 4 °C overnight in PBS containing 4% Triton X-100 and 2% normal goat serum (NGS). Thereafter, brains were washed in PBT for 1 h and counterstained with the primary monoclonal antibody nc82, which labels the pre-synaptic active zone protein, Bruchpilot. Brains were incubated in mouse nc82 antibody (DSHB, 1:50) mix containing 2% NGS in PBT for 2 nights at 4 °C. Thereafter, brains were washed in PBT for 1 h at room temperature and incubated in Cy3 goat anti-mouse (Jackson ImmunoResearch, 1:200) mix containing 2% NGS in PBT for 2 nights at 4 °C.

## Cell dissociation and counting

Following immunostaining, the brains were rinsed in PBT and transferred into a standard solution containing 40 mM sodium citrate and 1% triton X-100. The sample was heated at 60˚C for 10 min to soften the tissue. Sample was collected by centrifugation at 14000 g for 3 min, and the dissociation solution was carefully pipetted-out without disturbing the tissue. Mechanical dissociation was achieved by gently grinding the brain with a micropestle (Fisher

Scientific Cat# 12141364) inside a 1.5ml Eppendorf tube (**Fig 1B**). This procedure is expected to lyse the cell membrane and preserve the nuclear membrane. The tip of the micropestle was rinsed with 10 µl PBS. The 10µl volume was determined empirically to ensure suitability for counting.

Complete dissociation of the nuclei in PBS was ensured by pipetting up and down continuously for 30 sec before transferring onto the Neubauer counting chamber or hemocytometer (Hausser Scientific, Cat# 0267151B). The entire 10 µl homogenate was loaded onto the hemocytometer and total cell count was determined by averaging the number of fluorescent cells found within the 4-corner large squares and the center large square. Stained nuclei were located using the 10X objective under a fluorescence microscope (Axiom Imager.D2) and counted at 400X magnification. Nuclei that are stained with DAPI but not anti-elav were considered non-neuronal. To avoid counting errors due to background staining or cell debri, we ensured that neuronal cells were also stained with DAPI (**Fig 1C**).

To reduce variability in nuclei counts, we pooled three brains together in the same tube, dissociated and averaged the total counts, and divided this value by 3 to obtain an estimate for a single brain. This approach reduced the coefficient of variation in *Drosophila* whole brain cell count from 24.79% (when starting the IF protocol with a single brain) to 7.85% (when starting the IF protocol with three brains). The conventional IF protocol involves tissue dissociation prior to antibody staining [13]. Given the miniature size of these insect brains, staining prior to dissociation minimizes cell loss and enables visual tracking of the brains during antibody staining and washing.

## Results

Using the isotropic fractionator method, we estimated the cellular composition of the vinegar fly and mosquito whole brains and its sub-regions (**Table 1**). We find that the adult *D. melanogaster* whole brain has $217 \pm 4 \times 10^3$ cells, which is significantly less (P <0.0001) when

**Table 1. Cellular composition of *Drosophila* and mosquito brains.**

|  | Whole brain | | Central brain | | Optic lobe | | Male | | Female | |
|---|---|---|---|---|---|---|---|---|---|---|
|  | Estimates | CV | Estimates | CV | Estimates | CV | Estimates | CV | Estimates | CV |
| ***Drosophila melanogaster*** | | | | | | | | | | |
| All cells ($10^3$) | 217.12 ± 4.55 | 0.08 | 108.66 ± 2.55 | 0.07 | 119.68 ± 2.87 | 0.07 | 220.17 ± 6.54 | 0.08 | 213.64 ± 6.56 | 0.08 |
| Neuronal cells ($10^3$) | 199.38 ± 3.40 | 0.06 | 101.14 ± 3.65 | 0.11 | 107.27 ± 2.72 | 0.08 | 199.17 ± 5.55 | 0.07 | 199.17 ± 4.08 | 0.07 |
| Non-Neuronal cells ($10^3$) | 17.74 ± 3.50 | 0.74 | 7.52 ± 2.15 | 0.86 | 12.42 ± 3.06 | 0.74 | 21.00 ± 4.49 | 0.53 | 12.27 ± 5.08 | 0.60 |
| ***Aedes aegypti*** | | | | | | | | | | |
| All cells ($10^3$) | 248.53 ± 4.05 | 0.06 | 125.22 ± 2.56 | 0.07 | 121.39 ± 4.19 | 0.11 | 242.67 ± 4.30 | 0.05 | 255.24 ± 6.57 | 0.07 |
| Neuronal cells ($10^3$) | 217.91 ± 6.18 | 0.11 | 109.14 ± 3.55 | 0.12 | 112.00 ± 4.28 | 0.13 | 216.42 ± 8.48 | 0.10 | 219.62 ± 9.69 | 0.11 |
| Non-Neuronal cells ($10^3$) | 30.62 ± 4.38 | 0.54 | 16.08 ± 2.94 | 0.63 | 9.39 ± 2.23 | 0.79 | 26.25 ± 5.06 | 0.51 | 35.62 ± 7.35 | 0.51 |
| ***Anopheles coluzzii*** | | | | | | | | | | |
| All cells ($10^3$) | 253.04 ± 2.43 | 0.04 | 125.40 ± 2.65 | 0.06 | 121.62 ± 3.94 | 0.12 | 252.25 ± 3.15 | 0.03 | 251.75 ± 4.24 | 0.05 |
| Neuronal cells ($10^3$) | 223.02 ± 4.65 | 0.08 | 105.13 ± 3.67 | 0.11 | 107.14 ± 3.09 | 0.10 | 222.88 ± 5.80 | 0.07 | 223.17 ± 7.70 | 0.09 |
| Non-Neuronal cells ($10^3$) | 30.02 ± 4.26 | 0.55 | 20.27 ± 2.69 | 0.40 | 14.48 ± 3.41 | 0.85 | 20.78 ± 7.35 | 0.66 | 28.58 ± 6.04 | 0.56 |
| ***Culex quinquefasciatus*** | | | | | | | | | | |
| All cells ($10^3$) | 254.18 ± 3.77 | 0.06 | 134.13 ± 6.62 | 0.16 | 118.89 ± 2.78 | 0.74 | 253.83 ± 5.21 | 0.06 | 254.57 ± 5.91 | 0.06 |
| Neuronal cells ($10^3$) | 225.91 ± 7.22 | 0.12 | 108.53 ± 7.99 | 0.18 | 110.67 ± 3.95 | 0.11 | 234.00 ± 6.06 | 0.07 | 236.67 ± 3.61 | 0.17 |
| Non-Neuronal cells ($10^3$) | 28.27 ± 6.75 | 0.92 | 25.60 ± 5.98 | 0.74 | 8.2 ± 2.21 | 0.85 | 19.83 ± 5.06 | 0.52 | 25.4 ± 2.21 | 0.72 |

Values are mean ± SEM (standard error mean); Coefficient of variation (CV) = standard deviation/mean; total number of cells ($10^3$).

compared to the number of whole brain cells in *Ae. aegypti* (248 ± 4 x $10^3$), *An. coluzzii* (253 ± 2 x $10^3$), and *C. quinquefasciatus* (254 ± 4 x $10^3$). The neuronal cell population constitutes 91.8% of the entire *Drosophila* brain count. In mosquitoes, we recorded 87.7%, 88.1% and 88.9% neuronal brain cell population for *Ae. aegypti*, *An. coluzzii* and *C. quinquefasciatus* respectively. The number of neurons in the whole brain of the three mosquito species is significantly more than *Drosophila* brain neurons (P = 0.0048).

We further investigated the cellular composition of the insects' whole brain regions by dissecting it into two main parts: the central brain and the optic lobes. The number of DAPI-stained nuclei in the *Drosophila* central brain is significantly less than in the mosquitoes (P<0.0001). In the central brain, we counted a total of 108 ± 3 x $10^3$ cells for *D. melanogaster*, 125 ± 3 x $10^3$ cells for *Ae. aegypti*, 125 ± 3 x $10^3$ cells for *An. coluzzii*, and 134 ± 7 x $10^3$ cells for *C. quinquefasciatus* (**Table 1**). Unlike the central brain counts, the total number of DAPI-positive cells in the optic lobe for *D. melanogaster* (119 ± 3 x $10^3$) is not statistically different (P = 0.918) from *Ae. aegypti* (121 ± 4 x $10^3$), *An. coluzzii* (121 ± 4 x $10^3$) and *C. quinquefasciatus* (118 ± 3 x $10^3$).

We note that the number of nuclei counted by IF for the whole brain did not necessarily match the sum of the IF counts for the two optic lobes and the central brain (**Table 1**). This observation largely reflects technical limits on the precision or accuracy of the IF technique as counting was performed independently on different animals. We cannot rule out the possibility of additional tissue loss during the process of dissection, which could alter the tallying of the total cell number. The estimated percentage mean absolute error recorded in the DAPI-positive cells in the whole brain vs. the sum of the brain regions (optic lobes and central brain) was 5.17% (*D. melanogaster*), 0.77% (*Ae aegypti*), 2.38% (*An. coluzzii*), and 0.46% (*C. quinquefasciatus*).

Interestingly, comparisons of the neuronal cell population in the central brain and optic lobes were not significantly different for the four insect species (P = 0.723); these two brain regions contained approximately 100,000 neuronal cells (see S1 Fig for side-by-side comparison). However, non-neuronal cell population in the whole brain of mosquito species statistically exceeded that of *Drosophila* (P<0.0001; **Table 1**). In all the insect species examined, the DAPI-positive cell population was statistically greater (P<0.05) than the neuronal cells. The number of neuronal cells also exceeded the non-neuronal cells in the whole brain and its subregions (**Fig 2A–2D**). However, in the optic lobes of *C. quinquefasciatus*, the neuronal cell count was not significantly different from the DAPI-positive cells in this brain region (**Fig 2D**; P = 0.307), suggesting few non-neuronal cells present in the *C. quinquefasciatus* optic lobe. We observed low co-efficient of variation (CV) in the total cell count for DAPI-positive and elav-positive cells (**Table 1**). However, we recorded high CV values for non-neuronal cell population due to the greater level of dispersion around the mean count. This was not completely surprising given the small proportion of the non-neuronal cells (~10–15%) that constituted the insect nervous system [23, 25], which reduced the chance of obtaining consistent counts around the mean.

We next used IF to determine if there was sexual dimorphism in the cell population and cell types of the selected dipterans. At the level of analyses possible by IF, there was no statistical difference in the number of whole brain cells for male and female *D. melanogaster* (**Fig 2A**; P = 0.516), *Ae. aegypti* (**Fig 2B**; P = 0.119), *An. coluzzii* (**Fig 2C**; P = 0.463), and *C. quinquefasciatus* (**Fig 2D**; P = 0.463) when analyzed by independent sample T-Test. Similarly for cell types, the number of neurons and non-neuronal cells in the whole brain of male vs. female was not significantly different for *D. melanogaster* (P = 0.475 vs. P = 0.169), *Ae. aegypti* (P = 0.403 vs. P = 0.302), *An. coluzzii* (P = 0.488 vs. P = 0.467), and *C. quinquefasciatus* (P = 0.123 vs. P = 0.096). The IF counts trended to a greater number of non-neuronal cells in the whole

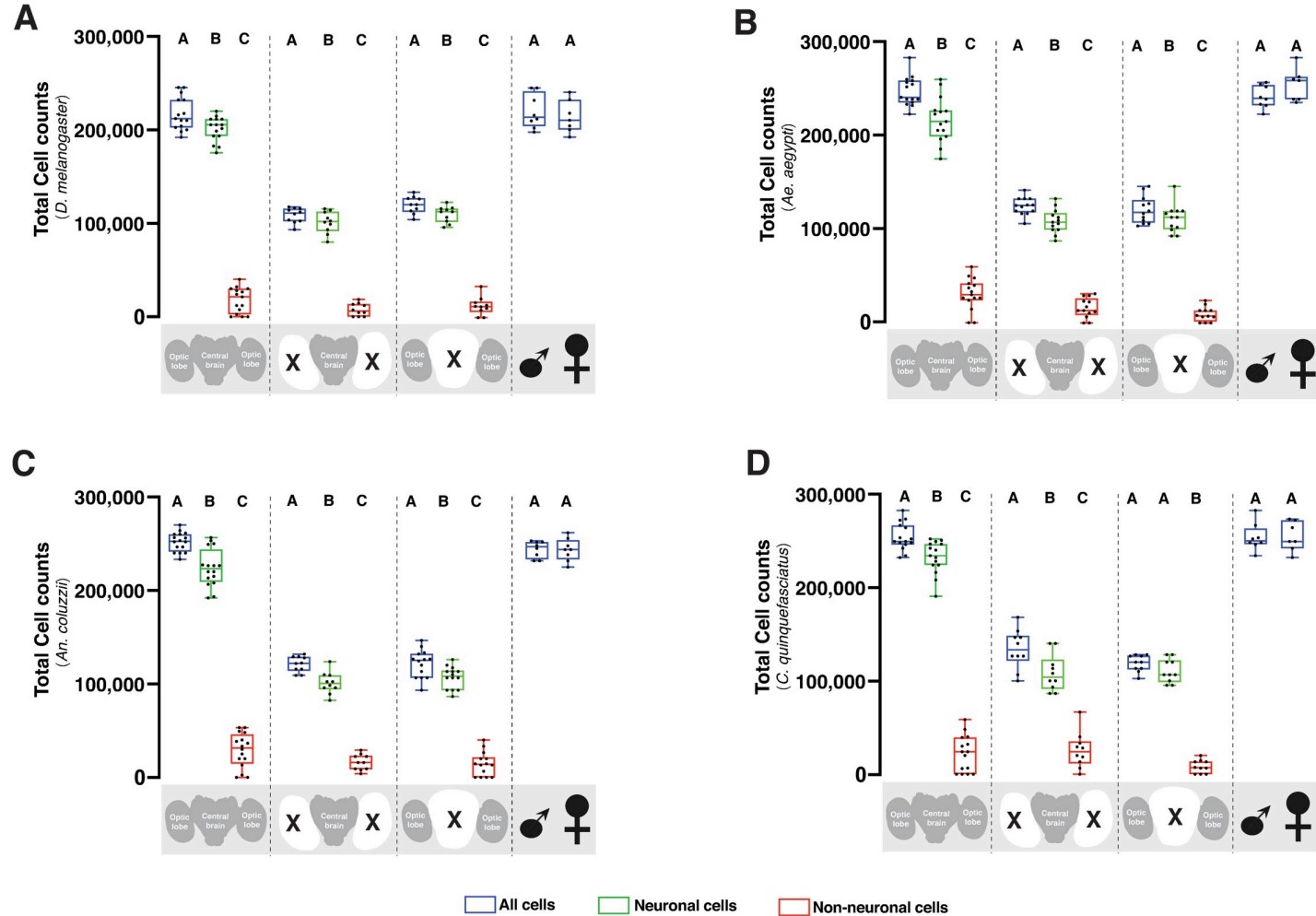

**Fig 2. Quantification of the cell population in vinegar fly and mosquito brains.** Box plot showing total cell counts, neuronal, and non-neuronal cell population in the whole brain, central brain, and optic lobe of *D. melanogaster* (2A), *Ae. aegypti* (2B), *An. coluzzii* (2C), and *C. quiquefasciatus* (2D). Blue plots represent all cells, while green and red plots represent neuronal and non-neuronal cells, respectively. Different groups (whole brain, central brain, optic lobes, and sex) are separated by dashed lines. Each data point represents an average of three brains. The number of trials for whole brain (n = 15), central brain (n = 10), optic lobes (n = 10), male (n = 8), and female (n = 7) is consistent for all animals. Within each group, different letters are statistically different (P < 0.05) when analyzed by one-way ANOVA with Tukey's post hoc test. Statistical comparisons between male and female total cell was performed by T-Test. Data analysis was performed using the GraphPad Prism 9 software package (GraphPad Software, San Diego, CA, USA).

brain of male *Drosophila* compared to the female (**Table 1**). However, this data was not statistically different (P = 0.083) when analyzed by T-Test.

## Discussion

We have used the isotropic fractionator method to provide experimental evidence for the number of neurons in adult *Drosophila* and mosquito brains. In this study, we estimated that the *Drosophila* brain contains just under 200,000 neurons, with roughly half of those neurons originating from the optic lobes. A study that examined the cellular composition of the *Drosophila* first instar larval brain estimated ~9,000 cells when counted on a hemocytometer [24]. We estimated over 200,000 cells in the adult *Drosophila* whole brain; this provides evidence for exponential increase in cell proliferation during brain development. The mosquito brain contains approximately 220,000 neurons, also with approximately half originating from the optic

lobes. The non-neuronal population of the brain, likely consisting mainly of glia, is approximately 18,000 cells in *Drosophila* and 31,000 cells in mosquitoes.

Although the Culicine (*Aedes* and *Culex* species) and Anopheline (*Anopheles* species) mosquitoes diverged about 120 million years ago [27, 28], their brain cell population is not statistically different by IF. The similarities in the brain cell population and cell types suggest that *Ae. aegypti*, *An. coluzzii* and *C. quinquefasciatus* brains are fundamentally similar at the cellular level. However, there were approximately 20,000 more cells in the mosquito whole brain compared to *Drosophila*. Fossil records and molecular studies posit that the ancestors of *Drosophila* and the subfamily Anophelinae diverged ~259.9 million years ago [28]. Future studies might be able to determine if particular neuronal regions in the mosquito brain have expanded in comparison to *Drosophila*, which might implicate neuronal specialization for certain behavioral tasks.

What can we derive from estimating the brain neuron numbers of these dipterans? The total number of neurons in an insect brain represents one measure of its computational power [29]. Neuronal cell population is crucial for brain functionality but is not the only determinant of brain information processing capacity. Other factors, including the number of neuronal connections and packing density, axonal conduction velocity, diversification of neuronal types and their properties [30, 31], all contribute to the broad behavioral repertoire seen in insects. In mammals, neuronal and non-neuronal cell population varies across every hierarchical order and follows a predictable scaling equation [32]. In mammalian brains, enrichment of neuronal populations correlate with the complex behaviors the brains can process [32]. From an evolutionary perspective, body size and brain size are not sufficient to explain the functional complexity of every species [1]. For example, an elephant's brain is ~3-fold larger than the brain of a human. However, the number of neurons estimated in a human cerebral cortex is triple that of an elephant [33]. This suggests that the number of brain neurons, but not necessarily brain volume or size, is a key determinant of brain functional complexity.

Sex difference in the number of neurons and non-neuronal cells has been reported in the human olfactory bulb, with females having significantly greater number of neurons and glial cells than males [5]. Sexually dimorphic neural circuitry has also been identified in the olfactory system of *Drosophila* [34] and mosquitoes [35, 36], which attests to the existence of this phenomenon in the sensory system of insects. However, the IF method could not identify sexual dimorphism in the brain cell counts of these insects. This does not necessarily rule out sexual dimorphism between brains, yet, does suggest that any sexual differences present would be at small populations or in neuronal processes or synaptic connections undetectable by this method. The emergence of new tools and methods to label smaller brain populations will aid in the identification of cellular differences, if any, in the insect brain.

IF is a simple method that can be used on any dissected tissue [13]. In this study, we optimized IF for estimating brain cell counts in insects with small brains. The required step of homogenization makes it impossible to reveal information about the structural arrangements of cells within the brain. However, it is a appropriate method to estimate cell populations. The IF method could be applied to a range of insect brains, allowing for better experimental comparison among a variety of organisms. For example, cellular differences could be investigated in the brain of social insects that display sexual dimorphism such as honeybees.

In addition, IF could be combined with immune or genetic labeling of cells to estimate the number of brain subpopulations. The relative ease of the method might even allow IF experiments to be incorporated into lab classroom curriculums. IF could also be applied in insect developmental studies to compare cellular differences in the developmental stages. For example, an estimate of neuronal proliferation occurring along developmental time points can be investigated. Studies using the *Drosophila* model to investigate human neurodegenerative

diseases [37, 38] may adopt IF to estimate neuronal population changes in the nervous system of aging *Drosophila*.

An accurate estimate of neuron numbers can set the expectations in complexity for full brain reconstruction [7, 39]. Unlike the EM-reconstruction performed on a single *Drosophila* female brain [7, 39], the IF approach allows the averaging of many brains to reduce variation among individuals. The future completion of the *Drosophila* whole brain EM-reconstruction project will provide an exact count of the number of cells in a single fly brain, and serve as an independent measure for the accuracy of IF in estimating cell numbers as reported in this study.

## Supporting information

**S1 Fig. Neuronal counts for vinegar fly and mosquito brains.** Data in Fig 2 were re-plotted to allow side-by-side comparisons of neuronal counts in (A) whole brain, (B) central brain, and (C) optic lobe amongst the four insect species.
(EPS)

## Acknowledgments

The following reagent was provided by the Centers for Disease Control and Prevention for distribution by BEI Resources, NIAID, NIH: *Culex quinquefasciatus*, Strain JHB, Eggs, NR-43025.

## Author Contributions

**Conceptualization:** Joshua I. Raji, Christopher J. Potter.

**Formal analysis:** Joshua I. Raji.

**Funding acquisition:** Christopher J. Potter.

**Investigation:** Joshua I. Raji.

**Methodology:** Joshua I. Raji.

**Supervision:** Christopher J. Potter.

**Visualization:** Joshua I. Raji.

**Writing – original draft:** Joshua I. Raji.

**Writing – review & editing:** Joshua I. Raji, Christopher J. Potter.

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
