## [Decision Letter · Decision Letter 0]

6 Apr 2021

The Number of Neurons in Drosophila and Mosquito Brains

PONE-D-21-06555

Dear Dr. Potter,

We’re pleased to inform you that your manuscript has been judged scientifically suitable for publication and will be formally accepted for publication once it meets all outstanding technical requirements. We nonetheless recommend that you implement the minor suggestions made by the reviewers to improve the manuscript. In particular, including a figure panel that compares neuron numbers across species would be a great addition to the work. 

Kind regards,

Matthieu Louis

Academic Editor

PLOS ONE

Additional Editor Comments (optional):

Reviewers' comments:

Reviewer's Responses to Questions

**Comments to the Author**

1. Is the manuscript technically sound, and do the data support the conclusions?

Reviewer #1: Yes

Reviewer #2: Yes

2. Has the statistical analysis been performed appropriately and rigorously? 

Reviewer #1: Yes

Reviewer #2: Yes

3. Have the authors made all data underlying the findings in their manuscript fully available?

Reviewer #1: Yes

Reviewer #2: Yes

4. Is the manuscript presented in an intelligible fashion and written in standard English?

Reviewer #1: Yes

Reviewer #2: No

5. Review Comments to the Author

Reviewer #1: In this paper, Raji et al. use isometric fractionator (IF) and immunofluorescence to determine the number of cells in the brains of fruit fly D. melanogaster and three mosquito species ( A. aegypti, A. coluzzii, and C. quinquesfasciatus). These are species with drastically different behavioral response that are governed by the brain. Even though anecdotal reports on the number of neurons in the fruit fly brain, the experiments determining the total number of cells in the brains of these 4 insects are lacking. Using antibody stainings for neuronal (anti-elav) and non-neuronal (anti-repo) cells followed by IF, the authors determine the number of neuronal and nonneuronal cell populations. These experiments identified approximately 200,000 cells in fruit fly brain. They also find that the number of cells is increased in mosquito populations as approximately 218,000, 223,000, and 226,000 for A. aegypti, A. coluzzii, and C. quinquesfasciatus, respectively. In each species approximately 90% of the brain cells were neuronal and rest were non-neuronal cells, without any overt sexual dimorphism between males and females in the total number of cells. This paper is well written and addresses a critical gap in experimental demonstration of number of neurons in insect species commonly used in laboratory research. The paper is suitable as is for publication in PlosONE. A few minor points:

1- Page 13 line 241 and 242: “ the Drosophila brain contains just under 200,000 neurons….”. This reads as if it is referring to a different study (maybe unpublished), not the findings in this paper. Where is this number coming from? Is there a way to cite it? Everyone mentions this number but I have always wondered where it came from as well. I think it was even present before the hemibrain connectome, but I also remember people in the field mentioning 100,000 neurons in the central brain.

2- Referring to the statement in page 21 line 268: To make the conclusion that states brain size and computational power has increased in mosquitoes compared to fruit flies, there needs to be some ratio analysis between the total cells in the body versus the brain. Is there any information on the total number of cells or at least the volume of the body to brain in flies and mosquitoes?

3- The authors mention that immunohistochemistry and IF method was unable to determine sexual dimorphism in any of the brains analyzed in this study. As the authors indicate, there might be methodological limitations that allow for the detection of small differences in the number of neurons between males and females. These are definitely known for the Drosophila brain where certain neuronal clusters in the courtship circuits show male and female specific differences in the number of neurons. Are there known sexual dimorphism in the brains of the mosquito species analyzed? It would be good to add this information to the discussion if it exists.

4- Also looking at the immunofluorescence images, it seems like cells in C. quinquesfasciatus brain seem to be slightly bigger. Is this accurate? Do the brain cells in each species have comparable sizes?

Reviewer #2: The authors isotropic fractionation with immunohistochemistry to estimate the total number of neurons and non-neuronal cells in the brains of Drosophila melanogaster and three disease vector mosquitoes. The find similar numbers of neurons in all four species, but a modestly increased number of non-neuronal cells in the three mosquitoes. Finally, they observed no significant differences in cell/neuron number between the brains of males and females of any species.

I think the methods are sound and the replication adequate. I agree with the authors that most references to neuron numbers in insects seems anecdotal. It is useful to have these direct empirical estimates.

I would suggest including a figure panel that compares neuron numbers across species. This seems to be the most interesting comparison, but it is never depicted visually and I found the text a little confusing with regard to whether there were any significant differences among species. If I understand correctly, there were no significant species variation in neuron counts, but there were differences in non-neuronal cell counts. A simple figure showing this result would remove any ambiguity.

My only other significant critique is that the text is confusing in places and contains many minor grammatical errors (e.g. inconsistent verb tense, mixing of singular and plural etc.). I provide several examples below. Most issues occur in the early parts of the text. I found the discussion to be overall very clear and interesting.

Minor points:

Abstract: Meaning unclear in the sentence, “Each insect brain consisted of 89% ± 2% neurons of the total cell population.” Do this mean that neurons made of 89% of all brain cells?

Line 119: I assume the authors mean Drosophila nc82 antibody rather than mouse?

Line 141-142: The authors wrote, “To reduce variability in nuclei counts, we pooled three brains and averaged the total counts to obtain an estimate for a single brain.” It’s not totally clear to me what was done. Three different brains were dissociated and assessed individually, then the numbers were averaged? Or three brains were pooled in the same tube and dissociated together. Then neuronal counts were divided by 3? If the latter, it would also be useful to report here the total number of ‘batches’ of 3 brains that were analyzed for each species. This information is in the Fig. 2 legend, but could be added to methods.

Line 191: The authors write, “Interestingly, comparisons of the neuronal cell population in the central brain and optic lobes were not significantly different for all the insects (P=0.723), approximately 100,000 cells.” The meaning is not clear here. Was the comparison between central brain and optic lobe within species? Or was it between species? What has approximately 100,000 cells?

Sentence starting on line 202 is also unclear.

Line 250 – Please explain which mosquitoes fall into which of the two subfamilies (e.g. that Aedes and Culex are Culicines and Anopheles are Anophelines)

Lin 266 – Sentence unclear: “Mammals that exhibit enriched neuronal populations correlate with the functional complexity offered by these brains [30].”

6. PLOS authors have the option to publish the peer review history of their article (what does this mean?). If published, this will include your full peer review and any attached files.

Reviewer #1: No

Reviewer #2: No

---

## [Editor Report · Acceptance letter]

27 Apr 2021

PONE-D-21-06555 

The Number of Neurons in *Drosophila* and Mosquito Brains 

Dear Dr. Potter:

I'm pleased to inform you that your manuscript has been deemed suitable for publication in PLOS ONE. Congratulations! Your manuscript is now with our production department. 

Kind regards, 

on behalf of

Dr Matthieu Louis 

Academic Editor

PLOS ONE